# Boosting the Photocatalytic Ability of TiO_2_ Nanosheet Arrays for MicroRNA-155 Photoelectrochemical Biosensing by Titanium Carbide MXene Quantum Dots

**DOI:** 10.3390/nano12203557

**Published:** 2022-10-11

**Authors:** Bingdong Yan, Zike Cheng, Caiyan Lai, Bin Qiao, Run Yuan, Chide Zhang, Hua Pei, Jinchun Tu, Qiang Wu

**Affiliations:** 1State Key Laboratory of Marine Resource Utilization in South China Sea, School of Materials Science and Engineering, Hainan University, Haikou 570228, China; 2Department of Clinical Laboratory of the Second Affiliated Hospital, School of Tropical Medicine, Key Laboratory of Emergency and Trauma of Ministry of Education, Research Unit of Island Emergency Medicine, Chinese Academy of Medical Sciences (No. 2019RU013), Hainan Medical University, Haikou 571199, China

**Keywords:** PEC biosensor, Ti_3_C_2_T_x_ MXene QDs, TiO_2_ nanosheet arrays, type Ⅱ heterojunction, microRNA-155 detection

## Abstract

The electrodes of two-dimensional (2D) titanium dioxide (TiO_2_) nanosheet arrays were successfully fabricated for microRNA-155 detection. The (001) highly active crystal face was exposed to catalyze signaling molecules ascorbic acid (AA). Zero-dimensional (0D) titanium carbide quantum dots (Ti_3_C_2_T_x_ QDs) were modified to the electrode as co-catalysts and reduced the recombination rate of the charge carriers. Spectroscopic methods were used to determine the band structure of TiO_2_ and Ti_3_C_2_T_x_ QDs, showing that a type Ⅱ heterojunction was built between TiO_2_ and Ti_3_C_2_T_x_ QDs. Benefiting the advantages of materials, the sensing platform achieved excellent detection performance with a wide liner range, from 0.1 pM to 10 nM, and a low limit of detection of 25 fM (S/N = 3).

## 1. Introduction

The ultrasensitive, rapid, and accurate detection of microRNA is very meaningful for the early diagnosis and prevention of disease [1]. Research has shown that the aberrant expression of microRNA-155 in the human body can be regarded as a critical detection index for some diseases, such as B-cell lymphoma [2] and breast cancer [3]. However, microRNA-155 is expressed only at the DNA level and not at the protein level; therefore, detecting microRNA-155 by traditional methods for early warning is very difficult [4]. Photoelectrochemical (PEC) biosensing is now attracting extensive attention for sensing nucleic acid and other diagnostic markers because of its inherently low limit of detection and high sensitivity. Generally speaking, there are two important parts in PEC biosensing [5]: (i) the PEC biosensing active species (catalytic signaling molecule to generate the detection signal) and (ii) the biological recognition elements (which are in contact with the active species). Therefore, active materials are very important for photoelectrochemical biosensing.

TiO_2_ is one of the most charming candidates for PEC biosensing due to its outstanding chemical stability, biocompatibility, and accessibility [6]. Titanium dioxide nanomaterials have been widely used in biological monitoring [7,8]. Sadly, pristine TiO_2_ suffers from a high carrier recombination rate, which significantly hinders the signal generation and collection of PEC sensors [9]. Coupling TiO_2_ with other semiconductors can achieve spatial separation of the photogenerated charges [10]. Proper band alignment and electron trapping would increase the concentration and lifetime of the photogenerated charges, thereby improving the catalytic ability of the material [11,12,13]. For this purpose, the interface between the two materials needs to be rationally designed. In principle, the morphology and contacting pattern of active species have to be rationally considered to maximize the contact area while reducing the interfacial defects caused by lattice mismatch between the two phases. On the one hand, the optoelectronic properties of composite materials are closely related to the configuration between the materials. For instance, compared with other forms of allotropes (such as graphene and carbon nanotubes), 0D carbon materials (such as carbon quantum dots) exhibit unique optoelectronic properties when combined with TiO_2_. On the other hand, the charge behavior of the materials is different when the heterojunction is built on different exposed crystal planes because (i) the generation rates of the photogenerated carriers on different crystal planes are different, and (ii) different work functions of different crystal planes can change the direction of electron flow between the heterojunctions [14].

Since their discovery in 2011, MXene materials have come into the spotlight due to their chemical stability, rapid charge-transfer kinetics, and tight interfacial coupling. Quantum dots derived from 2D materials exhibit excellent properties as compared to their 2D counterparts, such as more abundant active edge sites, bandgap widening, and tunable physicochemical properties [15]. In addition, compared with the other QDs, Ti_3_C_2_T_x_ QDs possess more abundant surface hydrophilic groups (–O and –OH), making them connect tightly with photoactive supporters. Hence, the Ti_3_C_2_T_x_ QDs could be a good co-catalyst for boosting the performance of the PEC biosensor. Song et al. employed Ti_3_C_2_T_x_ QDs as a photoactive material to promote the performance of TiO_2_-based PEC sensing. 

Herein, a PEC biosensing platform was fabricated for microRNA-155 detection. Two-dimensional TiO_2_ NS arrays were selected as the sensing active substrate. The exposed (001) crystal face of TiO_2_ enables the material to have higher catalytic performance. The Ti_3_C_2_T_x_ QDs were used as a co-catalyst for the photocatalysis of ascorbic acid (AA) and to suppress the recombination of the charge carriers inside the electrode. Their appropriate energy-band structure enables them to form a type II heterojunction with TiO_2_ to achieve efficient separation of electrons and holes. The S9.6 antibody was used as the microRNA recognition unit to identify DNA–RNA hybrid duplexes, and alkaline phosphatase (ALP) served as the catalytic signal generation unit. With reasonable material selection and interface design, excellent sensing performance can be expected.

## 2. Experimental Section

### 2.1. Synthesis of Ti_3_C_2_T_x_ MXene QDs

An amount of 1 g Ti_3_AlC_2_ was slowly added into 10 mL concentrated hydrofluoric acid solution (40 wt %). The mixture was stirred for 12 h to fully etch the aluminum atomic layer in Ti_3_AlC_2_. Afterward, the mixture was centrifuged until the pH was near neutral. After vacuum filtration, the sample was vacuum dried at 200 °C overnight. Then, 0.1 g of Ti_3_C_2_ powder was added to 10 mL of tetramethylammonium hydroxide (TMAOH, 1 wt%) and stirred for 12 h. The TMAOH-intercalated Ti_3_C_2_ powder was centrifuged at 8000 rpm, vacuum filtered, and vacuum dried at 200 °C. Finally, 50 mg of the sample was added to 10 mL solution of TMAOH (2.5wt%). The suspension was refluxed at 110 °C for a whole day, centrifuged at 12,000 rpm, and vacuum dried at 200 °C. 

### 2.2. Synthesis of Ti_3_C_2_T_x_ QDs/(001) TiO_2_/FTO Electrode

Initially, 10 mL of concentrated hydrochloric acid was mixed with equal amounts of deionized water to configure a dilute solution. Then, 385 μL of tetrabutyl titanate and 0.158 g of ammonium fluorotitanate were added into the mixture with constant stirring until a transparent color solution formed. The fluorine-doped tin oxide (FTO) substrates were ultrasonically cleaned with a glass cleaner and poured into a Teflon reaction kettle with a perforated Teflon base. After heating at 170 °C for 12 h, the FTO substrates were rinsed with water. The (001) exposed TiO_2_ NSs arrays were prepared after annealing in an air atmosphere at 450 °C for 3 h. Subsequently, Ti_3_C_2_T_x_ QDs (20 mg) were dispersed in 20 mL of water. In order to carry out the self-assembly process, the substrates were dropped vertically into the solution. The solution was placed in an oven at 50 ºC overnight. The Ti_3_C_2_T_x_ QDs slowly self–Organized onto the surface of TiO_2_ NSs with the volatilization of water. Finally, the Ti_3_C_2_T_x_ QDs/(001) TiO_2_/FTO electrodes were washed with ultra-pure water to remove the unconnected Ti_3_C_2_T_x_ QDs. 

### 2.3. PEC Detection of microRNA-155

To immobilize DNA, 20 μL of Au NPs (0.05 mg/mL) was added dropwise onto the Ti_3_C_2_T_x_ QDs/(001) TiO_2_/FTO surface. An amount of 20 μL of 0.5 μM probe DNA immobilization buffer was cast onto the Ti_3_C_2_T_x_ QDs/(001) TiO_2_/FTO electrode and incubated under humid condition for 12 h at 25 ºC. The electrode was denoted as DNA/Ti_3_C_2_T_x_ QDs/(001) TiO_2_/FTO electrode. The electrode was washed with a washing buffer and incubated with 20 μL of mercaptohexanol (MCH, 0.1 mM) for 1 h. Then, the DNA/Ti_3_C_2_T_x_ QDs/(001) TiO_2_/FTO electrode was incubated with 20 μL of different concentrations of microRNA-155 for 2 h. The RNA-DNA/Ti_3_C_2_T_x_ QDs/(001) TiO_2_/FTO electrode was washed with 0.1×SSC hybridization buffer to eliminate the unhybridized microRNA-155. Subsequently, 20 μL of the S9.6 antibody (20 μg/mL) was further incubated with the electrode for 1 h at 25 ºC in a humid cell. The S9.6-RNA-DNA/Ti_3_C_2_T_x_ QDs/(001) TiO_2_/FTO electrode was then washed with a buffer. Then, the electrode was incubated with 20 μL of IgG-ALP (25 μg/mL) at 37 °C for 1 h and keeping the surface moist. Finally, the PEC response of the ALP-IgG/antibody/RNA-DNA/Ti_3_C_2_T_x_ QDs/(001) TiO_2_/FTO electrode was recorded in the detection buffer at 0V. 

## 3. Results and Discussion

### 3.1. Electrode Construction and Sensing Mechanism of PEC Sensor

As shown in Figure 1A, layered Ti_3_C_2_T_x_ MXene were fabricated by a top-down method by etching the Al atomic layer in Ti_3_AlC_2_ with HF. The Ti_3_C_2_T_x_ QDs were prepared by the reflux hydrothermal method with TMAOH as the intercalating agent. Using ammonium fluorotitanate as a seed, (001) TiO_2_ NSs were grown on FTO glass through the hydrolysis of titanate in an acidic solution (Figure 1B). Ti_3_C_2_T_x_ QDs and (001) TiO_2_ NSs were joined together by a self-assembly process. The microRNA-155 detection process is shown in Figure 1C. Au NPs served as the reagent of the immobilization matrix for the thiol modified probe DNA. MCH was used for end capping of the electrode surface. After probe DNA hybridization with target RNA, rigid DNA:RNA double helix hybrids were combined with the S9.6 antibody. Afterward, the immunoreaction between the IgG and S9.6 antibody would lead to alkaline phosphatase immobilization. The alkaline phosphatase on the electrode surface could catalyze phosphorylated ascorbic acid in the detection solution to generate electron donor ascorbic acid, thereby increasing the electrode photocurrent and realizing the quantitative analysis of target microRNA-155.

### 3.2. Morphology Characterization of Ti_3_C_2_T_x_ QDs/(001) TiO_2_/FTO Electrode

Atomic force microscopy was used to observe the topography and size of the quantum dots. From Appendix A, the average thickness of Ti_3_C_2_T_x_ QDs was about ~1.0 nm, indicating that they were mostly single layer. FESEM was used to study the morphology of the (001) TiO_2_ NSs. The TiO_2_ NSs with a side length of about 2 μm and a thickness of about 150 nm uniformly grew on the surface of FTO glass (Appendix A). The FESEM images of the Ti_3_C_2_T_x_ QDs/(001) TiO_2_ composite are provided in Figure 2a,b. Compared with pure TiO_2_ NSs, there were no significant changes in the morphology of the composite electrodes after loading with Ti_3_C_2_T_x_ QDs. Transmission electron microscopy (TEM) images were provided to characterize the crystal information of TiO_2_ NSs. The HRTEM image (Figure 2c insert, middle part) revealed (200) and (020) atomic planes with a lattice spacing of 0.19 nm and an interfacial angle of 90°. The bright, periodically arranged diffraction spots in selected-area electron diffraction (SAED, Figure 2c insert, top right-hand corner) patterns indicated that the TiO_2_ NSs prepared were a single crystal with excellent crystallinity [16]. Proofread with standard PDF cards, the main exposed crystal plane of TiO_2_ nanosheets was (001) [17]. The introduction of Ti_3_C_2_T_x_ QDs was further identified by TEM images. Compared with pure TiO_2_ NSs, many small scales (~10 nm) appeared on the Ti_3_C_2_T_x_ QDs/(001) TiO_2_ composite (Figure 2d). The HRTEM image of the Ti_3_C_2_T_x_ QDs/(001) TiO_2_ composite is presented in Figure 2e. The lattice fringes with widths of 0.19 and 0.21 nm can be assigned to the (200) plane of TiO_2_ and the (100) plane of Ti_3_C_2_T_x_ QDs. The elemental mapping dots of the Ti_3_C_2_T_x_ QDs/(001) TiO_2_ composite for Ti and O were dense and apparent (Figure 2f–i) because TiO_2_ was dominant in this composite, whereas those for C were relatively scarce and primarily found around the sheet edges, indicating that Ti_3_C_2_T_x_ QDs successfully combined with the (001) crystal plane of TiO_2_ NSs.

### 3.3. Composition Characterization of Ti_3_C_2_T_x_ QDs/(001) TiO_2_/FTO Electrode

XRD pattern, Fourier transform infrared (FTIR) spectroscopy, and XPS analyses were performed for electrode composition characterization. The XRD results in Figure 3a indicated that FTO had peaks at 26.58°, 33.77°, 37.77°, 51.76°, and 65.19°, consistent with SnO_2_ (JCPDS No. 46-1088) [18]. Meanwhile, the TiO_2_ NS arrays had diffraction peaks at 25.28°, 37.80°, 48.05°, and 55.06°, assigned to the anatase TiO_2_ diffraction peaks (JCPDS No. 21-1276). No distinct characteristic diffraction peak of Ti_3_C_2_T_x_ QDs was found in the Ti_3_C_2_T_x_ QDs/(001) TiO_2_ sample, which was due to the low crystallinity and low content of the Ti_3_C_2_T_x_ QDs in the composites [19]. To further determine the functional group information of Ti_3_C_2_T_x_ QDs and TiO_2_, the Fourier transform infrared spectroscopy (FTIR) spectra of TiO_2_ NSs, Ti_3_C_2_T_x_ QDs, and Ti_3_C_2_T_x_ QDs/(001) TiO_2_ were presented in Appendix A. The (001) TiO_2_ composite film had some characteristic peaks at 3439, 1633, 1380, and 1110 cm^−1^, which were assigned to surface hydroxyl groups and adsorbed oxygen. Compared with pure TiO_2_, two new peaks emerged at 561 and 613 cm^−1^ after self-assembly, and they can be assigned to Ti-C and Ti–O, respectively [20]. 

The chemical bonding and functional groups of (001) TiO_2_ and Ti_3_C_2_T_x_ QDs/(001) TiO_2_ composite were also investigated by XPS spectrum. In Figure 3b, the high-resolution spectrum of Ti 2p of (001) TiO_2_ revealed two peak components at 458.8 eV (2p_3/2_) and 464.4 eV (2p_1/2_). After loading Ti_3_C_2_T_x_ QDs, the peak components of Ti 2p_3/2_ and 2p_1/2_ centered from low binding energy to high binding energy were attributed to the Ti-C, Ti-X from substoichiometric TiC_x_ (x < 1) or Ti_3_AlC_2_, Ti^2+^ ions and Ti^4+^ ions, respectively [21]. The spectrum of O 1s had two peaks located at 530.98 and 529.83 eV (Figure 3c), which were assigned to the Ti–OH species and the lattice oxygen [Ti–O_6_] species. As for the O 1s XPS spectra after Ti_3_C_2_T_x_ QDs were loaded, two new peaks were found at 531.78 and 533.58 eV, ascribed to the Ti-C–OH and Ti-C–O species, demonstrating the surface groups of Ti_3_C_2_T_x_ QDs were O and −OH [22]. The C 1s of (001) TiO_2_ can be divided into three characteristic peak components located at 288.4 eV, 286.5 eV, and 284.7 eV, which can be assigned to O-C=O, C=O, and C-C. Compared with pure TiO_2_, the introduction of Ti_3_C_2_T_x_ QDs led to the appearance of two new characteristic peaks. The characteristic peak at 282.3 can be assigned to the Ti-C inside the Ti_3_C_2_T_x_ QDs. Interestingly, compared with (001) TiO_2_ composite, a new component appeared at 283.03 eV after the self-assembly process, which could be assigned to the C−Ti−O_x_ bonding at the interfaces between Ti_3_C_2_T_x_ QDs and (001) TiO_2_ (Figure 3d) [23]. We believe that the O and –OH on the surface of Ti_3_C_2_T_x_ may act as rivet sites to connect to the five coordinated titanium atoms in (001) of TiO_2_ and form an atomic-scale interfacial heterojunction between 0D Ti_3_C_2_T_x_ QDs and 2D TiO_2_ NSs. 

### 3.4. PEC Performance Characterization of Ti_3_C_2_T_x_ QDs/(001) TiO_2_/FTO Electrode

To evaluate the catalytic ability of the materials to catalyze AA, time-resolved current response curves were obtained in an aqueous O_2_-saturated PBS solution containing AA (0.1 M) under light irradiation (365 nm). In Figure 4a, the photoelectric response of TiO_2_ NSs significantly improved after loading Ti_3_C_2_T_x_ QDs. To explain the enhanced catalytic ability, the photoelectric properties of the catalysts were evaluated. Photoluminescence (PL) spectra were also obtained to reveal the recombination efficiency of the carriers. In general, fluorescence emission at 420 nm represents the recombination of free excitons inside a material, whereas fluorescence emission at 480 nm represents surface state-trapping recombination [18]. Compared with TiO_2_ NSs, the emission intensity of Ti_3_C_2_T_x_ QDs/(001) TiO_2_ electrodes significantly decreased in both ranges (Figure 4b). The reduced recombination rate of photogenerated carriers could supply sufficient holes to activate –OH on Ti_3_C_2_T_x_ QDs, thereby significantly promoting the formation of reactive species (·OH) during the photocatalytic redox reaction. Time-resolved photoluminescence (TRPL) spectroscopy was performed to survey the lifetime of the electrons in (001) TiO_2_ and Ti_3_C_2_T_x_ QDs/(001) TiO_2_ electrodes. The average lifetime (τ_ave_) for the (001) TiO_2_ and Ti_3_C_2_T_x_ QDs/(001) TiO_2_ electrodes was 2.14 ns and 3.73 ns (Figure 4c). The carrier density of the electrodes was also investigated by the Mott–Schottky Equation (1) [23].
(1)$$\frac{1}{C^{2}}=\frac{2}{\varepsilon_{0}\varepsilon eN_{D}}\left[\left(U_{S}-U_{\mathrm{FB}}\right)-\frac{k_{B}T}{e}\right]$$

The carrier density $N_{D}$ can be obtained from the slope of the linear region of the Mott–Schottky plots (Figure 4d) on the basis of Equation (2).
(2)$$N_{D}=-\left(\frac{2}{e\varepsilon \varepsilon_{0}}\right){\left(\frac{d\left(1/C^{2}\right)}{d\left(U_{S}\right)}\right)}^{-1}$$
where $N_{D}$ is the electron density, *e* is the element charge value, *ε* is the dielectric constant (48 for anatase), *ε*_0_ is the vacuum permittivity, *C* is the space charge capacitance, and *U_S_* is the applied potential. The calculated *N*_D_ for the (001) TiO_2_ and Ti_3_C_2_T_x_ QDs/(001) TiO_2_ electrodes were 4.04 × 10^18^ and 8.18 × 10^18^, respectively. The photoelectric property tests implied that the introduction of Ti_3_C_2_T_x_ QDs could reduce the recombination rate, prolong the lifetime, and increase the density of the carriers in the electrode, thereby improving the catalytic ability of the electrode.

### 3.5. Electron Transfer Mechanism of Ti_3_C_2_T_x_ QDs/(001) TiO_2_/FTO Electrode

Ultraviolet–visible diffuse reflection spectrum (UV-vis DRS) and ultraviolet photoelectron spectroscopy (UPS) were combined to study the band structure and interface electron states of Ti_3_C_2_T_x_ QDs and (001) TiO_2_ (Figure 5a–d). Appendix A depicts the optical bandgap (E_g_) of the (001) TiO_2_ and Ti_3_C_2_T_x_ QDs, as derived from the Tauc Equation (3).
(3)$${\left(\alpha h\nu \right)}^{n}=A\left(h\nu -\mathrm{Eg}\right)$$
where α is the absorption coefficient, h is the Planck constant, ν is the photon frequency, *n* = 1/2 is the indirect bandgap semiconductors, A is a constant, and E_g_ is the bandgap. The bandgaps of (001) TiO_2_, Ti_3_C_2_T_x_ QDs were obtained as 3.16 and 2.91 eV, respectively. The cutoff energies (E_cut off_) of (001) TiO_2_ and Ti_3_C_2_T_x_ QDs were obtained as 16.67 (Figure 5b) and 17.05 eV (Figure 5d) from the UPS spectra. Their work functions (W) were calculated to be 4.55 and 4.15 eV, respectively. The valence band maximum (VBM) of (001) TiO_2_ and Ti_3_C_2_T_x_ QDs were determined from the binding energy onset as 2.49 (Figure 5a) and 1.93 eV (Figure 5c), which were −7.04 and −6.08 eV. The conduction band minimum (CBM) positions were -3.88 and −2.77 eV, which is the VBM plus the optical bandgap.

The band structures and schematic of electrode electron transfer of (001) TiO_2_ and Ti_3_C_2_T_x_ QDs are shown in Figure 5e,f. A type Ⅱ heterojunction was built between TiO_2_ and Ti_3_C_2_T_x_ QDs (Figure 5e). Because the CBM and VBM of Ti_3_C_2_T_x_ QDs were more positive than those of (001) TiO_2_, the photogenerated electrons from the conduction band (CB) of Ti_3_C_2_T_x_ QDs flowed to the CB of (001) TiO_2_ due to the lower energy level (Figure 5f). Given that the single crystalline (001) TiO_2_ were grown in situ on conductive substrate, the photogenerated electrons on the (001) plane were rapidly transferred to the FTO electrode. These electrons would flow to the counter electrode. As for the hole in the valance band (VB) of (001) TiO_2_, it will be injected to the VB of Ti_3_C_2_T_x_ QDs to oxidize the (–OH) groups on the Ti_3_C_2_T_x_ QD surface into ·OH free radicals (–OH+ h^+^ (hv) →⋅OH). These surface hydroxyl radicals can be regarded as the active species, thereby greatly improving the photocatalytic ability of the material. When the recognition process is completed, the ALP on the electrode surface converts Ascorbic acid-2-phosphate (AAP) into electron donor AA, and these active species can catalyze the oxidation of AA to generate dehydroascorbic acid (DHA) and generate photocurrent at the same time.

### 3.6. MicroRNA-155 Analytical Performance 

The PEC response current of stepwise modified electrodes was presented to corroborate the electrode modification process. In Figure 6a, a stable PEC response was obtained after Ti_3_C_2_T_x_ QDs were coated on the (001) TiO_2_ NSs substrate (curve a). Afterward, the PEC response was further raised after Au NPs were loaded, probably due to the good conductivity of Au NPs (curve b). The PEC response current of electrodes dropped gradually with the introduction of probe DNA, MCH, microRNA-155, and S9.6 antibody (curve c-f). This may be due to the poor electrical conductivity of nucleic acid and protein structures. However, when IgG-ALP was introduced into the system, the photoelectric response current of the electrode greatly improved (curve g). This is because the alkaline phosphatase can catalyze AAP to generate electron donor AA to enhance the photoelectric response. Figure 6b illustrates the EIS spectra of stepwise modified electrodes. The Ti_3_C_2_T_x_ QDs/(001) TiO_2_ electrode shows a semicircle (curve a) in the high-frequency region relating to the electron transfer resistance. Then, the electron transfer resistance decreased significantly when AuNPs were loaded (curve b). However, the electron transfer resistance increased continuously after probe DNA immobilization (curve c), MCH blocking (curve d), and hybridization with microRNA-155 (curve e). This could be due to the electrostatic repulsion between the negative ions (phosphate and acetate) and the redox probe of Fe(CN)_6_^3−/4−^. Electron transfer resistance further successively increased after the electrodes were incubated with S9.6 (curve f) and IgG-ALP (curve g) because of the insulativity of the protein structure. To explore the impact of the concentration of S9.6 and ALP-IgG, a concentration parameter adjustment experiment was performed in Figure 6c,d. It can be seen that the change of the response current also increases with the increase in the concentration. When the concentration of S9.6 reaches 20 μg/mL, and the concentration of ALP-IgG reaches 25 μg/mL, the current change reaches the maximum.

The response currents of the PEC platform with various microRNA-155 concentrations were tested (Figure 7a). The response current (*I*) showed a logarithmic relationship with the microRNA-155 concentrations (*c*), and the regression equation was *I* = 1.25lg*c* + 8.05 (R^2^ = 0.9964) (Figure 7b). Moreover, according to the literature [13], the LOD was calculated as 3.0×σ/S = 0.025 pM, where σ is the standard deviation of five times blank tests, and S is the sensitivity. The stability of the PEC platform was studied by continuous scanning under periodic light irradiation. Based on the relative standard deviation (RSD = 0.59%) of the response current in Figure 7c, the detection platform we built is very stable. Furthermore, the selectivity of the PEC platform was investigated by performing an anti-interference test with 1 nM microRNA-141, microRNA-121, and microRNA-21 as interferents. It can be seen that the response current of the detection platform to the interference is much smaller than that of the target, indicating that the detection platform has good anti-interference performance (Figure 7d). The performance of the detection platform is compared with the reported articles in Table 1. 

## 4. Conclusions

In this article, arrays of titanium dioxide nanosheets with a highly active (001) crystal plane were successfully prepared for microRNA-155 PEC detection. Zero-dimensional Ti_3_C_2_T_x_ QDs were successfully synthesized and used in titanium dioxide. The excellent performance was related to the higher surface energy due to the exposed (001) facet on TiO_2_ nanosheets. The better separation ability of the photogenerated carriers was due to the Ti_3_C_2_T_x_ QDs/TiO_2_ type Ⅱ heterostructure being able to reduce the loss of electron transfer inside the electrode. The faster electron transport caused by the 0D/2D nanostructure and lattice connection at the interface between Ti_3_C_2_T_x_ and TiO_2_ allowed the electrons generated by the detection to be collected more smoothly. The PEC sensor comprising the Ti_3_C_2_T_x_ QDs/(001) TiO_2_ electrode exhibited high stability, sensitivity, and selectivity for microRNA-155 detection.

## Figures and Tables

**Figure 1 nanomaterials-12-03557-f001:**
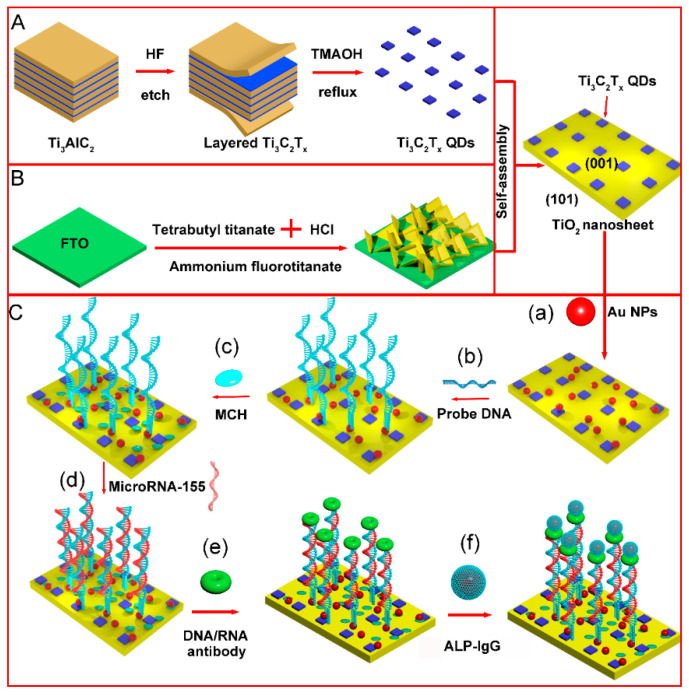
Schematic of PEC electrode construction and detection mechanism of microRNA-155 (**A**) synthesis process of Ti_3_C_2_T_x_ QDs (**B**) synthesis process of (001) TiO_2_/FTO electrode (**C**) detection process of microRNA-155 (**a**) dropping AuNPs (**b**) probe DNA loading (**c**) incubation with MCH (**d**) incubation with microRNA-155 (**e**) incubation with antibody (**f**) incubation with IgG-ALP.

**Figure 2 nanomaterials-12-03557-f002:**
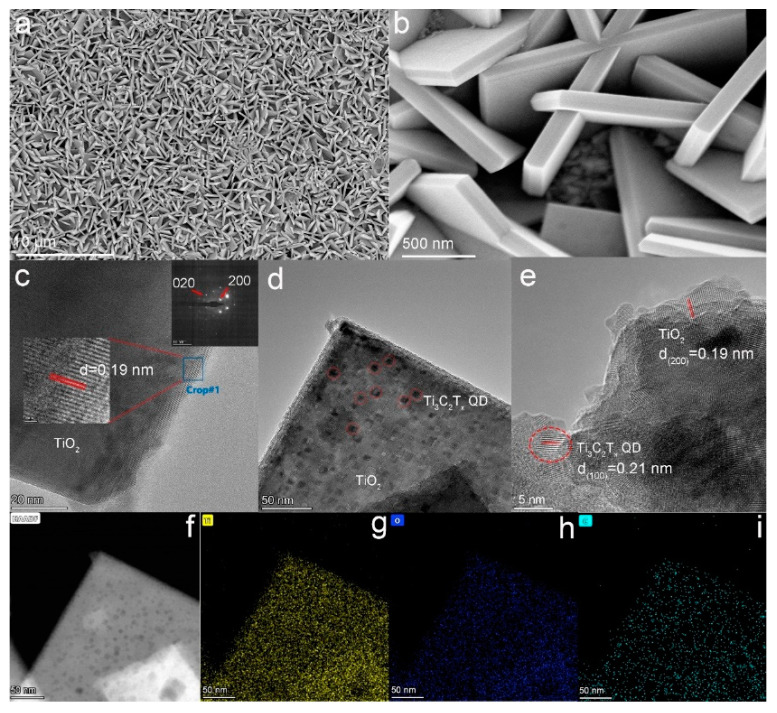
(**a**,**b**) FESEM of Ti_3_C_2_T_x_ QDs/(001) TiO_2_; TEM of (**c**) (001) TiO_2_ inset (middle part, HRTEM image, top right-hand corner, SAED) and (**d**) Ti_3_C_2_T_x_ QDs/(001) TiO_2_; (**e**) HRTEM of Ti_3_C_2_T_x_ QDs/(001) TiO_2_; and (**f**–**i**) mapping of Ti_3_C_2_T_x_ QDs/(001) TiO_2_ ((**g**): Yellow, titanium; (**h**): blue, oxygen; (**i**): cyan, carbon).

**Figure 3 nanomaterials-12-03557-f003:**
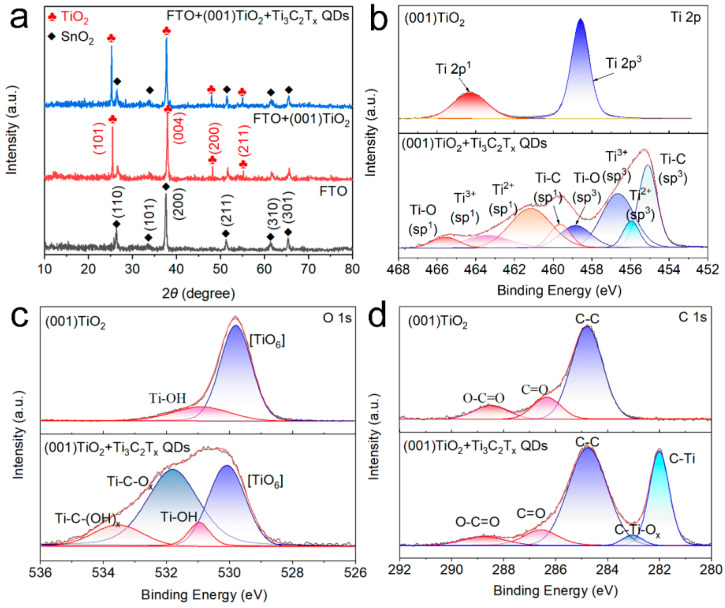
(**a**) X-ray diffraction (XRD) pattern of FTO glass, (001) TiO_2_ and Ti_3_C_2_T_x_ QDs/(001) TiO_2_ and X-ray photoelectron spectroscopy (XPS) results of (**b**) Ti 2p, (**c**) O 1s, and (**d**) C 1s orbital of (001) TiO_2_ and Ti_3_C_2_T_x_ QDs/(001) TiO_2_.

**Figure 4 nanomaterials-12-03557-f004:**
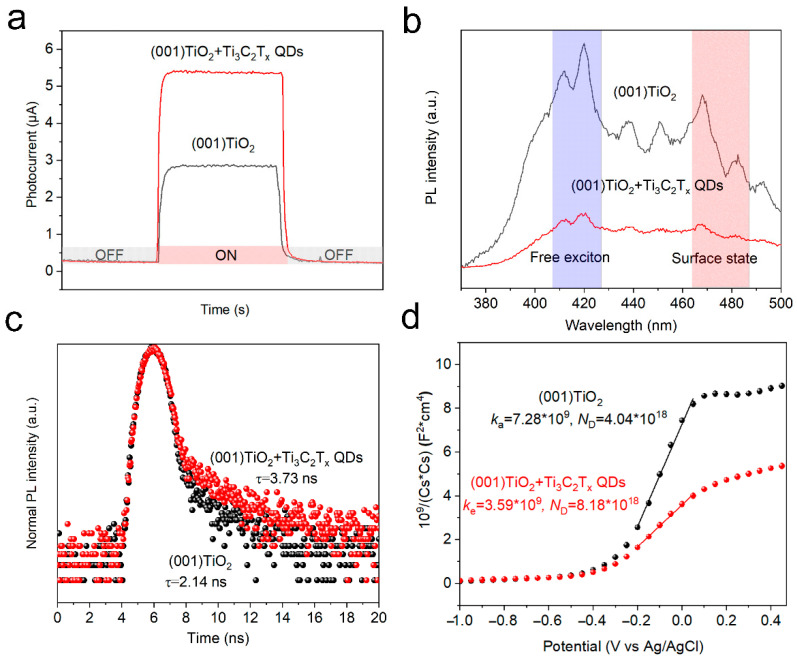
(**a**) Time-resolved current response curves, (**b**) Photoluminescence (PL) spectroscopy, (**c**) Time-resolved photoluminescence (TRPL) emission spectroscopy spectra, and (**d**) Mott–Schottky plot of pristine (001) TiO_2_ and Ti_3_C_2_T_x_ QDs/(001) TiO_2_ electrode.

**Figure 5 nanomaterials-12-03557-f005:**
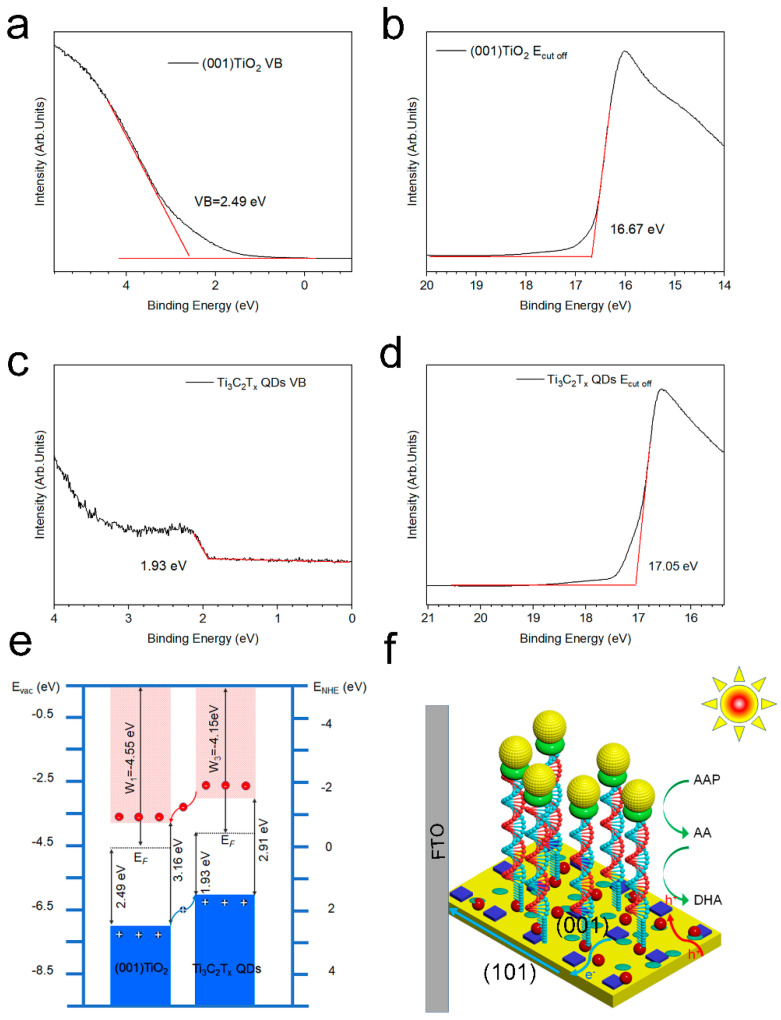
Ultraviolet photoelectron spectra: valence band spectra of (**a**) (001) TiO_2_ and (**b**)Ti_3_C_2_T_x_ QDs, cutoff energies spectra of (**c**) (001) TiO_2_ and (**d**)Ti_3_C_2_T_x_ QDs (**e**) band structure of (001) TiO_2_ and Ti_3_C_2_T_x_ QDs, and (**f**) schematic of electrode electron transfer.

**Figure 6 nanomaterials-12-03557-f006:**
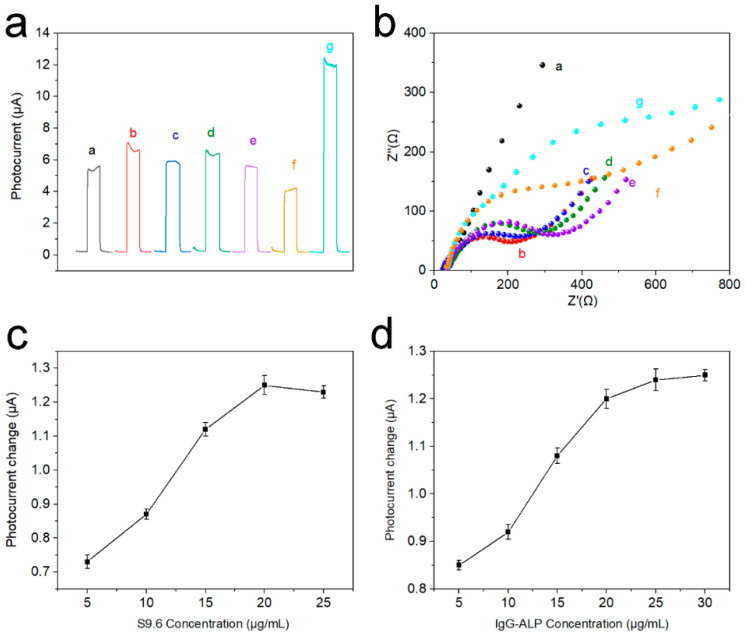
(**a**) Photocurrent response in detection buffer and (**b**) EIS plot in 5.0 mM Fe(CN)_6_^3−/4−^ solution of different electrodes: curve a, Ti_3_C_2_T_x_ QDs/(001) TiO_2_/FTO; curve b, after dropping AuNPs; curve c, after probe DNA loading; curve d, after incubation with MCH; curve e, after incubation with 1 nM microRNA-155; curve f, after incubation with antibody; curve g, incubation with IgG-ALP, (**c**) the change of the response current with different concentrations of S9.6 and (**d**) the change of the response current with different concentrations of IgG-ALP.

**Figure 7 nanomaterials-12-03557-f007:**
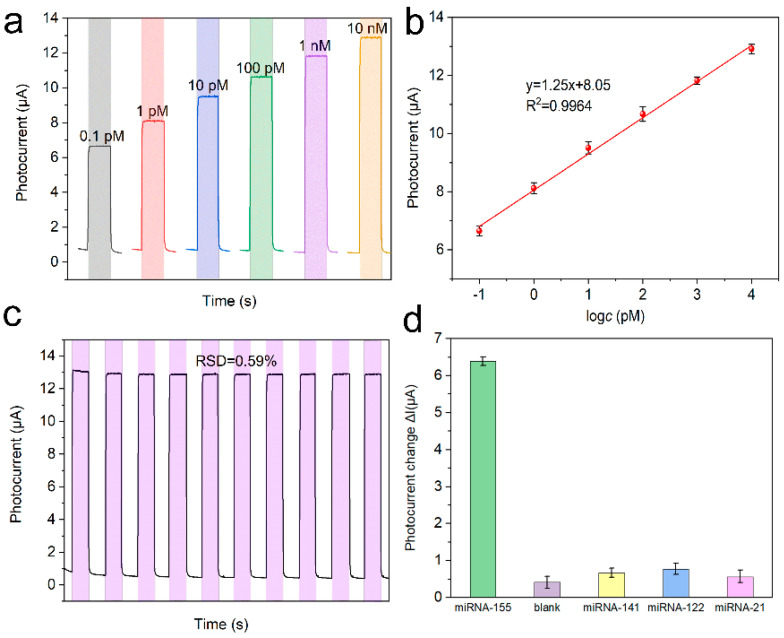
(**a**) Photocurrent response in detection buffer of the biosensor with different microRNA-155 concentrations (**b**) calibration curve, (**c**) stability of the PEC biosensor with 1 nM microRNA-155, and (**d**) selectivity of the PEC microRNA-155 biosensor with 1 nM different microRNAs.

**Table 1 nanomaterials-12-03557-t001:** Analytical performance of several microRNA-155 biosensors.

Biosensors	Dynamic Range (M)	Detection Limit (M)	References
CV	5.6 × 10^−12^–5.6 × 10^−5^	1.87 × 10^−12^	[24]
ECL	1 × 10^−12^–1 × 10^−5^	6.7 × 10^−13^	[25]
ECL	1 × 10^−14^–1 × 10^−5^	1.6 × 10^−10^	[26]
PEC	1 × 10^−11^–2 × 10^−8^	5 × 10^−12^	[27]
PEC	1 × 10^−13^–1 × 10^−8^	3.3 × 10^−13^	[28]
PEC	1 × 10^−13^–1 × 10^−8^	2.5 × 10^−13^	This work

## Data Availability

Data available in a publicly accessible repository. The data presented in this study are openly available in [FigShare] at [10.6084/m9.figshare.21291828], reference number [10.1000/data.20120401].

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
