# Peer review of "Boosting the Photocatalytic Ability of TiO2 Nanosheet Arrays for MicroRNA-155 Photoelectrochemical Biosensing by Titanium Carbide MXene Quantum Dots"

_nanomaterials, 2022, doi:10.3390/nano12203557_

Round 1
Reviewer 1 Report
The paper reports on nano-composite photo-electrodes for the detection of MicroRNA-155 . The paper can be of interest and it requires only minor revisions. The English style and form should be revised by a native speaker since some sentences and phrases are incorrect and difficult to understand.
The authors should pay more attention to figure sketches and plots. The captions should provide an extensive description of the figure, not only a one-row statement. For example, the caption of Figure 1 should briefly describe all the steps of fabrication and the working principle. Moreover, in the sketch of Figure 1 the last step is wrongly described as ALP-IgG-AuNPS whereas, accordingly to the description in the text, it should be ALP-IgG-AA. The same holds for Figure 2 caption. Please, report in the caption the association color-element for panels f-i. Please, correct the caption of Figure 6 where it is not clear to what are referred the steps a-f. In the calculation of the LOD, it is not clear how much is the value of S nor how it has been calculated. Please, report the value of S together with its error. In Figure 7C the response curve seems to be very regular, so why the RDS is so low? Please, clarify this point. The Conclusions paragraph should be improved. The authors claim about immobilization of 0D Ti3C2Tx QDs on TiO2 NSs whereas they only deposit the QDs on the NSs without any functionalization procedure. Stability of the sensor against pH and temperature variation should also be proven.
Reviewer 2 Report
The paper reports the fabrication of electrodes of two-dimensional (2D) titanium dioxide (TiO2) nanosheet arrays for microRNA-155 detection. The active crystal face is exposed to catalyze signaling molecules ascorbic acid. Zero-dimensional titanium carbide quantum dots were modified to the electrode and reduced the recombination rate of charge carriers.
Spectroscopic methods were used to determine the band structure of TiO2 and Ti3C2Tx QDs.
The nanosheet arrays presents good performances, with a wide liner range from 0.1 pM to 10 nM, and low limit of detection of 25 fM.
However, before publication I recommend to make few improvements:
- The inset writing from Fig. 5, 7 is too small. Make it more visible.
- I recommend to add short discussion and references about the state of art about TiO2 nano-structured materials used for bio-detection: (i) see paper http://dx.doi.org/10.1016/j.jallcom.2016.12.099 (ii) see paper or https://doi.org/10.1016/j.snb.2021.129843.
After minor revisions the paper is suitable for publication.
